# Protocol for a qualitative study to explore acceptability, barriers and facilitators of the implementation of new teleophthalmology technologies between community optometry practices and hospital eye services

Ann Blandford ,[1] Sarah Abdi ,[1] Angela Aristidou,[2] Josie Carmichael,[1,3] Giulia Cappellaro,[2,4] Rima Hussain,[3,5] Konstantinos Balaskas[3,5]

For numbered affiliations see end of article.

**Correspondence to**
Professor Ann Blandford;
a.blandford@ucl.ac.uk

## ABSTRACT

**Introduction** Novel teleophthalmology technologies have the potential to reduce unnecessary and inaccurate referrals between community optometry practices and hospital eye services and as a result improve patients' access to appropriate and timely eye care. However, little is known about the acceptability and facilitators and barriers to the implementations of these technologies in real life.

**Methods and analysis** A theoretically informed, qualitative study will explore patients' and healthcare professionals' perspectives on teleophthalmology and Artificial Intelligence Decision Support System models of care. A combination of situated observations in community optometry practices and hospital eye services, semistructured qualitative interviews with patients and healthcare professionals and self-audiorecordings of healthcare professionals will be conducted. Participants will be purposively selected from 4 to 5 hospital eye services and 6–8 affiliated community optometry practices. The aim will be to recruit 30–36 patients and 30 healthcare professionals from hospital eye services and community optometry practices. All interviews will be audiorecorded, with participants' permission, and transcribed verbatim. Data from interviews, observations and self-audiorecordings will be analysed thematically and will be informed by normalisation process theory and an inductive approach.

**Ethics and dissemination** Ethical approval has been received from London-Bromley research ethics committee. Findings will be reported through academic journals and conferences in ophthalmology, health services research, management studies and human-computer interaction.

## INTRODUCTION

Ophthalmology is one of the busiest outpatient clinics in England, accounting for 8% of all hospital outpatients' attendances.[1] Most hospital eye services (HES) referrals originate from community optometrists

## STRENGTHS AND LIMITATIONS OF THIS STUDY

⇒ This study forms part of a large multicentre study (The HERMES study) that will collectively provide real-world evidence on the implementation of novel teleophthalmology technologies.
⇒ A key strength of this study is analysing the facilitators and barriers of the implementation of novel teleophthalmology technologies from the perspectives of multiple stakeholders including patients and primary and secondary eye care professionals.
⇒ Another strength of this study is using multiple methods (observations, interviews, self-audiorecording) to collect data from multiple hospital eye services and affiliated community optometry practices in England.
⇒ One limitation of this study is that most interviews will be conducted via videoconferencing or telephone, limiting the researcher's ability to build rapport with the interviewees.

(CO) in high street optician practices, who are the main providers of primary eye care in the UK.[2] Retinal disorders (eg, macular pathologies, retinal vascular pathologies and suspected retinal tears/detachments) are the most referred conditions.[3] The growing use of optical coherence tomography technology (OCT) in CO practices is believed to have contributed to the increase in retinal referrals to hospitals.[4 5] OCT is a non-invasive scanning technology that generates high-resolution, three-dimensional images of the retina.[6] OCT has transformed ophthalmology practice in the last decade, leading to better detection and understanding of common retinal conditions such as age-related macular degeneration (AMD).[6] However, the success of this technology in improving retinal care

for patients may have been limited by the referral process between CO and HES. Unnecessary and inaccurate referrals, rereferrals from CO and deficits in replies from HES are common issues in the referral process, increasing the burden on secondary care and, consequently, delaying access to timely eye care for patients who need it.[2 3] Therefore, there is an urgent need to explore potential solutions to improve the referral process and manage patient flow between CO and HES.

Teleophthalmology has emerged as a viable alternative to delivering eye care that may improve patients' access to timely and appropriate care.[7–9] Teleophthalmology is a means to provide ophthalmic care at a distance using information and communication technology.[8 10] A variety of eye care delivery models have been reported to benefit from teleophthalmology. For example, Caffrey et al[8] identified 62 discrete models of care that can be improved by teleophthalmology, including eye screening, patient consultations, emergency services, supervision of procedures, staff training and remote supervision. In the referral process, teleophthalmology services typically involve primary healthcare professionals (eg, community-based optometrists) obtaining images (eg, OCT, slit-lamp or retinal imaging) and transmitting them via an electronic system to secondary care.[8] A secondary care ophthalmologist then reviews these images and decides on the management of the case, which might involve meeting the patient, remotely monitoring them or continuing their management in primary care.[8 11] Teleophthalmology can have several benefits in the context of triage. For example, in one scoping review, teleophthalmology was found to contribute to reducing face-to-face appointments with ophthalmologists by 16%–48% through reducing inappropriate and unnecessary referrals.[7] Similarly, implementing remote retinal imaging-based referrals reduced the waiting time for patients to see an ophthalmologist from 14 weeks to 4 weeks.[7] Teleophthalmology has been found to improve elderly patients' access to specialist eye care and reduce workload on specialist centres and unnecessary visits.[10] Patients also reported high levels of satisfaction with teleophthalmology services due to reduced cost and time of travel, as well as increased accessibility to services.[11] A recent systematic review has also emphasised the potential of teleophthalmology to serve as an alternative eye care delivery model by demonstrating its feasibility and cost-effectiveness for the management of various eye conditions in several countries including the UK.[12] Additionally, in recent years, advances in artificial intelligence (AI), particularly in deep learning, hold great promise for expanding the use of teleophthalmology.[13–16] Deep learning can improve referrals by identifying patients who are more likely to develop a specific condition and require urgent care or frequent follow-ups, increasing patients' access to appropriate eye care.[13 15] Several recent studies have demonstrated comparable performances of deep learning algorithms to experts in diagnosing different eye conditions.[14 17 18] For instance, in one study, a deep learning algorithm reached or exceeded experts' performance in assessing urgent referrals from two independent sets of OCT scans (n=997, n=116) for a range of retinal conditions.[17] Similarly, the accuracy of a deep learning algorithm to assess AMD from fundus images has been found to range between 88.4% and 91.6% compared with human experts.[18]

However, despite these promising findings, triaging referrals via teleophthalmology has been limited in practice. For example, during the COVID-19 pandemic, a period associated with increased adoption of telehealth applications,[19] primary care optometrists were less willing to adopt teleophthalmology in the context of referrals.[20] Although the study did not explore in depth reasons for this limited adoption, this finding is not surprising. Generally, implementing digital health interventions in practice is acknowledged to be complex due to the multiple components that should be considered during implementation.[21–24] These include professionals' and patients' acceptance of the technology, staff training and education, changes in staff roles and practices, the organisational culture, capacity and readiness to accept innovations, and the wider context (eg, policy and regulations).[22 24] The application of deep learning algorithms in ophthalmology referrals also brings with it a new set of challenges. For example, there are risks related to data security and privacy, as well as potential harm from false negative diagnosis that may impact the implementation and acceptance of deep learning algorithms for clinical image classification.[14–16]

Deep learning algorithms are also characterised by a lack of transparency or explainability, sometimes referred to as the 'black box' phenomenon, which makes it difficult for healthcare professionals and patients to understand how they reached their output.[14 15 17] This raises the question of whether health professionals and patients would trust the use of a 'black box' for referrals.[17] Most work to increase the explainability of AI models has focused on the development of post hoc explanations of outputs, using methods such as saliency maps. However, these explanations are based on limited access to the 'inner workings' of models and have been criticised for a lack of stability, as well as for failing tests of utility and robustness.[25] To address post hoc short-comings, self-explaining AI, whereby complex interpretable models are built bottom up, has been proposed and developed. These produce explanations that are intrinsic to the model while still maintaining a high performance.[26 27] Overall, recent evidence suggests that teleophthalmology and AI decision support tools have the potential to improve the referral process between CO and HES. However, to improve the uptake of these technologies in practice, it is important to identify the factors that facilitate or hinder their implementation.

## Aims and objectives

Previous research on facilitators and barriers of teleophthalmology implementation has mainly focused on

diabetic retinopathy screening,[28–30] with limited research focusing on facilitators and barriers in the referral process between CO and HES on other retinal conditions. Therefore, this study aims to assess patients' and healthcare professionals' acceptance of, and barriers and enablers for, the adoption of two innovative digital technologies supporting referral pathways between CO and HES. These are a teleophthalmology platform and the Moorfields-DeepMind AI Decision Support System (DSS). A human–computer interaction (HCI) approach will be used in this study, to understand professionals' and patients' interactions with the proposed technological solutions as well as the contexts in which these technologies will be implemented. Five research objectives address the overall aim of this study:

1. To understand current workflows and practices of staff and patients in CO and HES so as to identify key user requirements for teleophthalmology tools from the perspectives of both groups.
2. To understand workflows and practices of staff and patients in CO practices and HES with already established teleophthalmology pathways to identify technical, logistical and human factors affecting implementation of teleophthalmology in practice.
3. To identify factors that shape professionals' and patients' attitudes to, and trust in, the Moorfields-DeepMind AI, and how to present information in ways that instil appropriate confidence.
4. To understand whether and how work practices are likely to change following the adoption of Moorfields-DeepMind AI.
5. To identify factors that ease the deployment of a digital referral platform to ensure acceptability and acceptance by all user groups, and to understand the adoption process.

## METHODS AND ANALYSIS
### The HERMES study
The current protocol focuses on the detailed design of the qualitative element of the 'Teleophthalmology-enabled and AI-ready referral pathway for CO referrals of retinal disease trial' (the HERMES study). HERMES is an interventional superiority cluster randomised trial that aims to compare standard practice for referral of suspected retinal diseases with a teleophthalmology digital link between CO and HES. A substudy will also be conducted as part of the trial that integrates the trial data to assess the diagnostic accuracy of an AI DSS (the Moorfields-DeepMind algorithm) for the automated referral recommendation for retinal disease. Detailed methods of the HERMES study are described elsewhere.[31] The qualitative research element presented in this paper will run across both studies to provide evidence on implementation.

### Study design and setting
A theoretically informed, qualitative study will be performed to explore patients' and healthcare professionals' perspectives on teleophthalmology models of care and AI DSS. A combination of situated observations with semistructured interviews with healthcare professionals, semistructured interviews with patients and self-audiorecording of healthcare professionals will be conducted. This approach will enable us to understand the contexts in which the two new technologies will be implemented, focusing on understanding workflows, practices and user requirements, as well as identifying potential barriers and facilitators to implementation. It will also enable us to gain an in-depth understanding of staff and patients' expectations and experiences with the implementation of the new technologies. The study will be conducted at 4–5 HESs and 6–8 affiliated optometry community practices. Data collection is planned to start in November 2021 and end in May 2022.

### Participant selection
#### Sampling
Purposive sampling will be applied to recruit participants who are representative of relevant patient and professional groups. This type of sampling is used to select participants who are most likely to produce valuable data.[32] Patient participants will be selected if they meet the following criteria:

► Able to communicate in English, understand the study and give informed consent.
► Adults (≥18 years) attending the involved CO practices who underwent an OCT scan.
► Individuals who in the opinion of the CO have any suspicion of a retinal condition (including dry AMD, wet AMD, diabetic retinopathy, macular oedema, macular holes, epiretinal membranes, central serous chorioretinopathy, genetic eye disease).

Patients with retinal conditions that are not routinely visualised or diagnosed using an OCT scan or those with conditions that prevent acquisition of good quality OCT will be excluded. This includes peripheral retinal comorbidities such as peripheral retinal degeneration, retinal tear, retinal detachment, peripheral retinochoroidal tumours, Coat's disease, Retinopathy of Prematurity, Familial Exudative Vitreoretinopathy, Sickle-cell retinopathy.

Professional participants will include CO and clinicians (medics or specialist optometrists) with a minimum of 2 years' experience of independent practice in retinal clinics in HESs. Some of the participants' characteristics (eg, their level of experience) will be monitored during recruitment to ensure that diverse views are included in the sample.

Participants will be recruited from three settings: (1) community optometry clinics in the control arm (pre-transitioning to teleophthalmology); (2) community optometry clinics in the intervention arm (post-transitioning to teleophthalmology) and (3) HESs. These settings will help us understand and compare experiences and work practices before and after implementing the new teleophthalmology technologies, as

well as identifying barriers and facilitators during their implementation. A total of 4–5 HESs and 6–8 community optometry practices (3–4 practices from the control arm and 3–4 practices from the intervention arm) will be included in the study.

For the observations, it is expected that valuable insight will be obtained from observing a total of 10–15 clinician-patient consultations (3–5 consultations in each setting). These numbers were estimated based on the research team's previous knowledge and experience on conducting observations in healthcare settings. However, insight from the first few observations will further inform the number of consultations required to achieve sufficient input from the observations.

For the interviews, the aim is to interview a total of 30–36 patients from 6 to 8 CO practices (5–6 patients from each participating CO) and up to 30 healthcare professionals (up to 10 in each setting, noting that many of the participating CO practices employ fewer than 5 optometrists). Data saturation, that is, no new information emerges from the sampled units, will also guide sample size.[33 34] Healthcare professionals in the intervention arm or post-transitioning to teleophthalmology should have sufficient experience with the teleophthalmology platform before participating in the interview. However, we don't have a specific period of exposure to the platform as the aim is to gain diverse views from practices at different stages of implementation.

For the self-audiorecording, the aim is to collect self-audiorecordings of referral decisions of participating healthcare professionals in CO and HES.

## Methods of approach
### Observations
The observations will focus on understanding general clinical practices and work routines. Thus, the observations might involve patients, but not specifically those with suspected retinal diseases. Managers of community optometry practices and secondary eye clinics will be approached to gain permission to conduct observations in their practices.

### Interviews
Two sets of interviews will be conducted.

A first set of interviews will focus on individuals with suspected retinal disease. Only patients who undergo an OCT and, in the opinion of the CO, have any suspicion of a retinal condition will be invited to participate in an interview. Potential patient participants will be invited to participate following their consultation at a participating CO practice. The optometrist will explain the study to potential participants, highlighting its purpose, possible advantages and disadvantages, and what it entails. Potential participants will be given sufficient time to think about their participation and ask questions about the study. The researcher will call potential participants to obtain their decision to participate and book a provisional interview date for those who agree to participate.

Interviews will be conducted at the optometry practice where the participant was recruited, or via telephone or video conferencing.

A second set of interviews will focus on professional participants at the HES and the community optometry practices, who will be invited to participate in interviews by the researcher. Interviews with professional participants will be conducted via video conferencing, or at the hospital or practice.

### Self-audiorecording
During the initial interview with healthcare professionals at the community optometry practices and the HES, participants will be invited to participate in the self-audiorecording data collection exercise described below.

## Data collection and analysis
### Theoretical approach
Most digital health interventions can be viewed as complex interventions as they include multiple components that interact at both individual and organisational levels.[21 23 35] The explicit use of a theoretical lens when evaluating the implementation of these interventions can enhance our understanding of factors that may influence their success or failure.[36 37] In this study, normalisation process theory (NPT) will be used as a theoretical lens in gathering and analysing the data. NPT is concerned with understanding and explaining factors that may facilitate or inhibit the incorporation of complex interventions into routine practice.[38 39] NPT focuses on understanding the work that individuals and groups need to do for a complex intervention to become 'normalised' and embedded in practice, particularly in a healthcare context.[39–42] Thus, a starting point of this theory is understanding current practices, that is, how people work and what they actually do.[40] NPT comprises four components that determine the normalisation of a complex intervention in practice.[39 40] These are: (1) coherence, which refers to participants' understanding of new technology and practices associated with it; (2) cognitive participation, which refers to the preparedness of participants to engage and use the technology; (3) collective action, which refers to the work that participants do to use the technology and (4) reflective monitoring, which refers to participants' appraisal of the new technology.[23 39 42] There is evidence for the stability and consistency of NPT constructs across various contexts, advocating their use to assess, describe or improve the implementation potential of complex interventions.[39 41 43] NPT has also been used to explore users', including patients' and healthcare professionals', expectations of digital health interventions as well as barriers and facilitators to engaging with these interventions,[37 42 44 45] although limited evidence is available on teleophthalmology and AI DSS. In this study, it is envisaged that the use of NPT will help better understand the implementation process of these two technologies in routine practice and identify factors that may contribute to a successful implementation.

## Design of observations, interviews and self-audiorecordings
### Observations
The aim of the observations is to gain familiarity with the contexts in which the two innovative technologies will be implemented. In particular, it will aim to establish an understanding of current practices and work routines, and identify any differences in the workflows between practices. This is an important step given that understanding what people do and how they work in real life is a core focus for NPT. Additionally, findings from the observations will help set the context for the semistructured interviews with healthcare professionals. The latter will then be used to have a more in-depth discussion with healthcare professionals regarding what would and would not work in practice which will help to identify the user requirements for the teleophthalmology platform.

Observations will be conducted in all settings (optometry practices and HESs), focusing on clinician–patient interactions around the diagnostic and referral process. Specifically, the researcher will take field notes on the workflow, how referral decisions are made and communicated to patients, the clinician interaction with the new teleophthalmology platform, and any facilitators or barriers experienced during the interaction. To facilitate capturing this data, the flow and sequence work models from contextual design will be used.[46] The flow model describes communication and coordination patterns that are important to accomplish the work, while the sequence model represents the detailed steps that people take to accomplish the tasks and the problems that they may encounter while doing them.[46] Detailed work model diagrams will be kept of all observations conducted in CO and HES.

### Interviews
The aim of the interviews is to gain an in-depth understanding of the expectations, perceptions, and experiences of patients and health professionals with the teleophthalmology platform. All interviews will be semistructured, allowing us to address the study aim, informed by NPT, while also following up on new insights as they emerge.[47] All professional participants will be interviewed once, with the option of participating in two further short interviews. The purpose of these additional interviews is to gain professionals' reflections on their propensity to adopt AI tools and to change their work practices following AI adoption. Two approaches will be used to conduct the semistructured interviews with healthcare professionals: contextual inquiry interviews and critical incident technique.

Contextual inquiry is a method commonly used in the HCI field to gain a deep understanding of users' work practices.[46 48] It is based on the premise that users are tacitly aware of their own work practices as they are immersed in their everyday activities.[46] To understand their actions and reveal their motivations, intents and strategies, it is important to observe and speak to them in the context in which they perform their day-to-day activities.[46] In other words, contextual inquiry involves conducting observations and following them up with questions to understand the work at hand.[47] In this study, contextual inquiry with healthcare professionals will complement the observations made in HESs and optometry practices.

Critical decision method (CDM), originated from the critical incident technique, is a cognitive task analysis approach used to elicit expert knowledge.[49] The CDM focuses on a retrospective analysis of critical incidents experienced by the interviewees.[50] In the context of HCI studies, critical incidents can include events when the technology failed or the system experienced particular demands.[47] The CDM uses a set of techniques to minimise recall biases and aid the interviewees to recall critical decisions as accurately as possible.[50] For example, the technique involves probing the interviewee to identify and describe a specific critical incident or incidents from beginning to end.[49] The researcher then composes a decision timeline and employs probe questions which allow the interviewee to provide corrections or more details.[49] The interviewee is also asked 'what-if?' questions to understand what might have happened differently. In this study, critical incident interviews will be conducted with healthcare professional participants in the intervention arm, to gain a deep understanding of their perceptions and experiences with the teleophthalmology platform as well as explore barriers to its implementation in practice (eg, when the platform failed and reasons for that).

A semistructured topic guide will be used in all interviews and will include questions related to the research topic and NPT. The topic guide will be tailored to each group (patients and healthcare professionals in the intervention and control arms) as well as to suit the approach employed (contextual inquiry and CDM). The interview procedure will follow the five steps to conduct HCI semistructured interviews.[47] Step 1 (opening the conversation) aims to put participants at ease and assure them they have the desired knowledge and expertise. Step 2 (introducing the research) aims to introduce the topic and ensure that participants are aware of the purpose, reaffirming their confidentiality and right to withdrawal, and requesting permission to record the interview. Step 3 (beginning the interview) aims to gain contextual information about the participant, such as their role, technology use and prior experience, which may help formulate the subsequent questions. Step 4 (during the interview) aims to gain in-depth information about the topic under investigation. NPT components (coherence, cognitive participation, collective action and reflective monitoring) will inform the questions in this step. Questions about coherence will focus on participants' expectations from the teleophthalmology platform, as well as its perceived benefits and barriers. Questions based on cognitive participation will explore participants' engagement with the teleophthalmology platform and the issues they may face when using this new technology. Questions about collective action will focus on participants' views on the impact of the teleophthalmology platform on eye care and practice, as

well as the changes that may be required to integrate this new technology in routine practices. Questions based on reflective monitoring will explore participants' perspectives on how the teleophthalmology platform should be implemented in the future. For the AI DSS, issues around the 'black box' phenomenon, as well as the optimal place in the care pathway, confidence and trust will be investigated. Probes such as anonymised screenshots from the digital referral platform and illustrative prototypes from the DeepMind algorithm will be used to support the exploration of the themes. Step 5 (closing the interview) will include ending the interview, providing the participant with an opportunity to express more thoughts, and thanking them for their contribution to the study and the design of the technology. All interviews will be audiorecorded, with participants' permission, and transcribed verbatim.

### Self-audiorecording

Self-audiorecording is a method with demonstrated scientific value for examining the decision processes of professionals.[51] The aim of the self-audiorecordings is to study whether and how exposure to the Moorfields-DeepMind AI referral decision changes the work practices of professionals in community optometry and HES.

Both community optometry and HES participants will be invited to record themselves (self-audiorecord) talking out loud about referral decisions. Self-audiorecordings will take place when healthcare professionals are alone (ie, after the patient has exited the room and without a researcher in the room). Following their self-recording, some healthcare professionals will be informed of the referral decision that the Moorfields-DeepMind AI DSS would make for the same patient, while others will not have this information. The allocation of participants in the groups will follow the allocation of the broader HERMES study. Participants will not be aware of which group they belong to when they first sign up for the study. Those healthcare professionals informed of the AI DSS referral decision will be further invited to record themselves talking out loud about the AI DSS referral decision and how it relates to the original human referral decision. The self-audiorecordings are not used to make an assessment of the referral but to understand how professionals make decisions as experts.

### Data analysis

Data gathering and analysis will be interleaved so that later data gathering is informed by findings from earlier analysis. A combination of inductive and deductive thematic analysis will be used to analyse data from the interviews, observations and self-audiorecordings, following Braun and Clarke's guidance on conducting a thematic analysis.[52] The analysis will start with familiarising oneself with the data early on by listening to audiotapes, reading transcripts and field notes. An open approach will be followed at the start of the coding, where data from the first few transcripts and field notes will be open-coded line-by-line, enabling interesting codes and insights to emerge from the data. Analysis will then be done deductively where codes will be informed by the research questions. In one analytical direction, codes will be informed by the NPT constructs (coherence, cognitive participation, collective action and reflective monitoring). In this direction, coding of the transcripts will be conducted independently by two researchers (SA and JC) with different backgrounds (ophthalmology, and digital health). SA and JC will meet fortnightly to discuss the codes and will resolve any disagreement by discussion. In a related analytical direction, coding will be conducted in a 'semigrounded theory' way,[53] whereby the researchers adopt established professional learning and development constructs in the coding process while still allowing for a change in the direction of enquiry during the analysis of the data. In this analytical direction, coding of transcripts will be conducted by two researchers (GC and AA) who will discuss fortnightly emerging insights with the broader research team. The coding scheme from interviews will inform the coding of self-recordings, for which we identify emerging themes and their evolution over time (per individual participant and per theme). Across both analytical directions, codes will be reviewed for similarities, differences and relationships and will be categorised into preliminary themes. These themes will be reviewed against the codes and coded text and will be organised into final themes. The wider research team will meet monthly to discuss the analysis, and the preliminary and final themes. NVivo V.20 software will be used to manage data analysis.

### Patient and public involvement

Eighteen patients were consulted during the preparation phase of the HERMES study. The consultation focused on patients' general perceptions of teleophthalmology, trust in technology and potential concerns about impersonal care or reduced opportunities to interact with healthcare professionals. Patients' perceptions of the central concept of the project was positive and patients recognised the potential benefits of teleophthalmology such as reducing waiting times and unnecessary visits to hospital. Several patients also emphasised the importance of providing information during attendance at community optometry practices around the pathways, the experience to be expected during their visit and timescale for obtaining feedback. Generally, patients' inputs reinforced the importance of introducing a comprehensive qualitative element to the study to capture patients' perceptions around digital models of eye care.

Additionally, the study is overseen by a steering committee including representatives of patients group. The steering committee will meet at least once a year with provision for additional meetings when input is required for potential protocol amendments or issues arising during the study. An end of study debrief is planned with all PPI contributors which will include discussions on the

prioritisation and dissemination of study results to both the public and relevant healthcare professionals.

## ETHICS AND DISSEMINATION

Health Research Authority and Health and Care Research Wales ethical approvals have been obtained from London-Bromley Research Ethics Committee (Rec ref number: 20/LO/1299). Participant information sheets will be provided to all potential participants. Written or audio/video recorded informed consent will be obtained from all participants before they participate in the study. All interviews will be conducted at a time and place convenient to participants. Participants will be reminded of their rights to withdrawal from the study without there being negative consequences on their work or the care they receive.

All data will be handled following the General Data Protection Regulations, UK data protection act 2018 and the Research Governance Framework for Health and Social Care. Participants' anonymity and confidentiality will be maintained during the study. Written informed consent forms will be stored in a locked cabinet in the principal researcher's office. Interviews will be conducted using encrypted audiorecorders and recordings will be removed from the portable device permanently as soon as they are transferred to an access-restricted folder on the university home drive. People transcribing the interviews will be subject to a nondisclosure agreement. Field notes and interview transcripts will be pseudonymised, which means that any personal information will be removed from the data before the analysis, and participants will only be identifiable using a study identification number. Pseudonymised data and the study identification log will be stored in two separate access-restricted folders on the University's home drive. Access to data will be restricted to the research team only.

Findings will be reported through academic journals and conferences in ophthalmology, health services research, management studies and HCI.

**Author affiliations**
[1]UCL Interaction Centre, University College London, London, UK
[2]School of Management, University College London, London, UK
[3]Moorfields Eye Hospital NHS Foundation Trust, London, UK
[4]Department of Social and Political Sciences, Bocconi University, Milano, Italy
[5]Institute of Ophthalmology, UCL, London, UK

**Contributors** AB and SA designed the study protocol. AA and KB contributed to the study design. SA prepared the first draft of the manuscript. AB, AA, JC, GC, RH and KB reviewed and contributed to subsequent drafts. All authors reviewed and approved the final draft of the manuscript.

**Funding** This work is supported by NIHR Health Technology Assessment grant number 18/182. AA is funded through the UKRI Future Leaders Fellowship research grant MR/S033009/1.

**Competing interests** None declared.

**Patient and public involvement** Patients and/or the public were involved in the design, or conduct, or reporting, or dissemination plans of this research. Refer to the Methods section for further details.

**Patient consent for publication** Not applicable.

**Ethics approval** Health Research Authority (HRA) and Health and Care Research Wales (HCRW) ethical approvals have been obtained from London-Bromley Research Ethics Committee (Rec ref number: 20/LO/1299).

**Provenance and peer review** Not commissioned; externally peer reviewed.

**ORCID iDs**
Ann Blandford http://orcid.org/0000-0002-3198-7122
Sarah Abdi http://orcid.org/0000-0002-4395-8257

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
