## [Reviewer comments · BMJ Open]

ARTICLE DETAILS

TITLE (PROVISIONAL)	Protocol for a qualitative study to explore acceptability, barriers, and facilitators of the implementation of new teleophthalmology technologies between community optometry practices and hospital eye services
AUTHORS	Blandford, Ann; Abdi, Sarah; Aristidou, Angela; Carmichael, Josie; Cappellaro, Giulia; Hussain, Rima; Balaskas, Konstantinos

VERSION 1 – REVIEW

REVIEWER	David Chen National University Hospital
REVIEW RETURNED	01-Feb-2022

GENERAL COMMENTS	The authors describe a detailed study protocol on evaluating factors and barriers on the implementation of two teleophthalmology technologies between community optometry (CO) practices and hospital eye services (HES). The two innovative technologies discussed are 1) teleophthalmology models of care and 2) Artificial Intelligence Decision Support System (AI DSS). This is a timely topic of significant relevance. Improved clarity on the individual evaluative components for the two innovative technologies would be appreciated. Major comments 1) While the protocol was written as an evaluation of both innovative technologies, the majority of the protocol and resource is allocated towards evaluation of AI DSS, while there is less emphasis on the evaluation of teleophthalmology in general. A more balanced approach would be appreciated. 2) Page 8, Lines 27 - 30: Please specify how "community optometry clinics in the intervention arm (post-transitioning to teleophthalmology)" is defined. Do these clinics all have a similar workflow / standardised practice? Is there a minimum duration for which these clinics would have transitioned to teleophthalmology for them to qualify? Minor comments 1) Page 8, Lines 27 - 30: Do all these hospital eye services already have existing workflows for teleophthalmology? I note they are affiliated with the optometry community practices (Page 7, Lines 42 - 44) 2) The observation segment is on general clinical practices and work routines which may not involve patient with suspected retinal diseases, though the objective would be to identify key user requirements for tele-ophthalmology (Page 6, Lines 28 - 32). Could
---

	the value-add of these observations be skewed by a disproportionate number of patients in CO who do not require such services (e.g. refractive errors and/or anterior segment conditions which would not require either AI DSS or teleophthalmology)? 3) Page 13, Lines 14 - 40: For the self-audio recording, please clarify whether healthcare professionals included are those from CO only (i.e. the ones making the referrals), or would include the ones from HES as well (i.e. the ones receiving the referrals) 4) Page 13, Lines 27 - 33: Please elaborate on how the allocation of participant into the two groups (would be made, and whether there is any specific ratio for allocation. If statistical calculation is performed, kindly elaborate on this as well 5) Page 14, Lines 8 - 31: Please specify how 'regularly' is defined in each of the circumstances mentioned (SA & JC, GC & AA, wide research team)
--	---

REVIEWER	Ji-Peng Li Moorfields Eye Hospital NHS Foundation Trust
REVIEW RETURNED	12-Feb-2022

GENERAL COMMENTS	worthwhile exercise to find out barriers to existing tech introduction can be more succinct
---

REVIEWER	Renoh Chalakkal University of Otago
REVIEW RETURNED	17-Feb-2022

GENERAL COMMENTS	The paper discusses different barriers and factors affecting the acceptability of a telemedicine system applied to ophthalmology specialization. The Paper is well written and has identified most of the relevant factors affecting the widespread use of such a portal. It is a very relevant topic in the current scenario where a majority of consultations are being done online. Please find specific comments below:  1. There are a few recent literature surveys conducted that try to identify how effective and widespread is the use of teleophthalmology in the pre-and post- pandemic times. Including these references can help readers to understand its importance. (https://journal.nzma.org.nz/journal-articles/teleophthalmology-in-the-post-coronavirus-era-open-access, https://www.ncbi.nlm.nih.gov/pmc/articles/PMC8500493/ , etc.) 2. The "black-box" problem of using deep learning methods in triaging has been recently addressed by self-explainable deep learning models. Would be better to include a brief detail about these developments in the introduction section (pg.5, 143-53) 3. Please expand the acronym HERMES. The paper might be of interest to professionals from engineering science who are not familiar with the medical terms/studies 4. Pg.8 112-14 -- Please elaborate on the know retinal co-morbidities/conditions in the exclusion criteria. Exclusion criteria need to be specific and self-explanatory. 5. Pg.8 141-43 -- How 10-15 clinician-patient consultations will be conducted? A brief detail regarding how this number was estimated would help. 6.
---

VERSION 1 – AUTHOR RESPONSE

No.	Reviewers' comments	Our response
	1st reviewer's comments	
	The authors describe a detailed study protocol on evaluating factors and barriers on the implementation of two teleophthalmology technologies between community optometry (CO) practices and hospital eye services (HES). The two innovative technologies discussed are 1) teleophthalmology models of care and 2) Artificial Intelligence Decision Support System (AI DSS). This is a timely topic of significant relevance. Improved clarity on the individual evaluative components for the two innovative technologies would be appreciated.	Thanks for your comment. We have clarified the point regarding evaluating the technologies. Please refer to our responses below.
	Major comments	
1	While the protocol was written as an evaluation of both innovative technologies, the majority of the protocol and resource is allocated towards evaluation of AI DSS, while there is less emphasis on the evaluation of teleophthalmology in general. A more balanced approach would be appreciated.	Thanks for your comment. Two of the study objectives aim to understand real-life implementation of the teleophthalmology platform in routine practice (Objective 1 & Objective 2, page 5, line 162-168). These objectives will be addressed by two qualitative elements which are situational observations and semi-structured interviews with healthcare professionals and patients. Most of the questions in the interview with HCPs and patients will focus on their expectations and experience with the teleophthalmology platform (page 12, line 385-393). Similarly, the aim of the observations is to understand the context in which the teleophthalmology platform will be implemented as well as to understand any issues faced by the healthcare professionals in interacting with this platform (page 10, line 322). However, we recognise that this might have not been very clear in the text, especially in the description of the interviews. We have made some minor corrections to the text to make it clearer (please refer to page 10, line 322, page 11, 335, page 12, line 385-393) With regards to AI DSS: some additional resources and methods (self-audio recordings &

		two additional interviews with HCPs) have been used for its evaluation; we believe that this is important as AI DSS is a more novel technology and more likely to cause controversy if implemented in the referral process compared to other teleophthalmology technologies.
2	Page 8, Lines 27 - 30: Please specify how "community optometry clinics in the intervention arm (post-transitioning to teleophthalmology)" is defined. Do these clinics all have a similar workflow / standardised practice? Is there a minimum duration for which these clinics would have transited to teleophthalmology for them to qualify?	We used the terms post-transitioning and pre-transitioning to differentiate between the two arms of the HERMES trial, i.e., the arm that implements the teleophthalmology platform (intervention arm or post-transitioning to teleophthalmology) and the arm that does not implement teleophthalmology (control arm or pre-transitioning to teleophthalmology). HCPs in the intervention arm or post-transitioning to teleophthalmology should have sufficient experience with the teleophthalmology platform before participating in the interview. However, we don't have a specific period of exposure to the platform as the aim is to gain diverse views from practices at different stages of implementation. Additionally, as mentioned in the protocol, we intend to conduct observations in practices in the intervention arm which will help us understand if there are any differences in the workflows and routines between these practices. These points have been clarified in the text, please refer to page 8, line 243-247, and page 10, line 312-313.
	Minor comments	
1	Page 8, Lines 27 - 30: Do all these hospital eye services already have existing workflows for teleophthalmology? I note they are affiliated with the optometry community practices (Page 7, Lines 42 - 44)	Participating Hospital Eye Services don't have existing teleophthalmology workflows at the onset of the study. These are established in each of the 4 participating Hospital Eye Services for the purposes of the HERMES study. Each participating HES will therefore have two referral workflows for community optometry referrals: their standard practice referral workflow for CO practices in the control arm and all other non-participating CO practices in their catchment area; and a teleophthalmology workflow specifically established as per the study protocol for handling referrals from CO practices randomised to the intervention arm.
2	The observation segment is on general clinical practices and work routines which may not involve patient with suspected retinal diseases, though the objective would be to identify key user requirements for tele-ophthalmology	We are aware that observations will not be limited to patients with suspected retinal conditions. Suspected cases of retinal conditions are usually identified during the consultation with the optometrist rather than beforehand, making it impractical to focus only on patients with

	(Page 6, Lines 28 - 32). Could the value-add of these observations be skewed by a disproportionate number of patients in CO who do not require such services (e.g. refractive errors and/or anterior segment conditions which would not require either AI DSS or teleophthalmology)?	suspected retinal conditions. Additionally, the aim of the observations is to establish an understanding of current work routines and practices, which will help to set the context for the semi-structured interviews with healthcare professionals. The latter will then be used to have a more in-depth discussion with healthcare professionals regarding what would and wouldn't work in practice which will help to identify the user requirements for the teleophthalmology platform. This has been clarified in the text, please refer to page 10, line 314-318
3	Page 13, Lines 14 - 40: For the self-audio recording, please clarify whether healthcare professionals included are those from CO only (i.e. the ones making the referrals), or would include the ones from HES as well (i.e. the ones receiving the referrals)	The self-audio recording will be performed by both CO and HES. We have clarified this on page 13, line 406.
4	Page 13, Lines 27 - 33: Please elaborate on how the allocation of participant into the two groups (would be made, and whether there is any specific ratio for allocation. If statistical calculation is performed, kindly elaborate on this as well	The allocation of participants in control and treatment groups will follow the allocation of the broader study. We have clarified this on page 13, line 412-413.
5	Page 14, Lines 8 - 31: Please specify how 'regularly' is defined in each of the circumstances mentioned (SA & JC, GC & AA, wide research team)	We have the clarified the frequency of the team meeting in text. Please refer to page 13-14, line 433, 439, 445.
	Reviewer 2 comments	Our response
	Reviewer's overall comment: Worthwhile exercise to find out barriers to existing tech introduction can be more succinct	Thanks for your feedback.
	Reviewer 3 comments	Our response
	Reviewer's overall feedback: The paper discusses different barriers and factors affecting the acceptability of a telemedicine system applied to ophthalmology specialization. The Paper is well written and has identified most of the relevant factors affecting the widespread use of such a portal. It is a very relevant topic in the current	Thanks for your feedback.

	scenario where a majority of consultations are being done online. Please find specific comments below:	
1	There are a few recent literature surveys conducted that try to identify how effective and widespread is the use of teleophthalmology in the pre-and post-pandemic times. Including these references can help readers to understand its importance.	Thanks for sharing the references. We have included the systematic review reference to emphasize the potential of teleophthalmology to support eye care delivery. Please refer to page 4, 105-108.
2	The "black-box" problem of using deep learning methods in triaging has been recently addressed by self-explainable deep learning models. Would be better to include a brief detail about these developments in the introduction section (pg.5, 143-53)	We have added some more information regarding the explainable deep learning models in the introduction. Please refer to page 4-5, line 138-145
3	Please expand the acronym HERMES. The paper might be of interest to professionals from engineering science who are not familiar with the medical terms/studies	The HERMES acronym has been expanded. Please refer to page 6, line 180-182.
4	Pg.8 112-14 -- Please elaborate on the known retinal co-morbidities/conditions in the exclusion criteria. Exclusion criteria need to be specific and self-explanatory	The objective of this exclusion criterion was to exclude pathologies that are not routinely visualised or diagnosed using an OCT scan i.e., non-macular pathologies. So the term 'retinal co-morbidities' should be interpreted as 'peripheral retinal co-morbidities' including: peripheral retinal degeneration, retinal tear, retinal detachment, peripheral retino-choroidal tumours, Coat's disease, Retinopathy of Prematurity, Familial Exudative Vitreoretinopathy, Sickle-cell retinopathy. All participating optometrists in the HERMES study are provided with clarifications on the interpretation of this exclusion criterion during study-specific training. This exclusion criterion has been further clarified in the manuscript. Please refer to page 7, line 215-219.
5	Pg.8 141-43 -- How 10-15 clinician-patient consultations will be conducted? A brief detail regarding how this number was estimated would help.	These numbers were estimated based on the research team's previous knowledge and experience on conducting observations in healthcare settings, with the aim of generating insight that serves as a baseline for the interviews with HCPs. However, insight from the first few observations will further inform the number of consultations required to achieve sufficient input from the observations. This has been clarified in

		text, please refer to page 7-8, line 234-238.
--	--	---